# Factors Affecting the Adoption of Online Database Systems for Learning among Students at Economics Universities in Vietnam

**Thi Minh Phuong Nguyen** 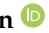

School of Accounting & Auditing, National Economics University, 207 Giai Phong Street, Hanoi 1400, Vietnam; phuongntm@neu.edu.vn

**Abstract:** This study aims to evaluate the determinants that influence the adoption of online databases in the learning process of students at economics universities in Vietnam. A quantitative study with a meta-analysis was conducted by utilizing structural equation modeling (SEM). The sample consisted of 492 students from economics universities located in Vietnam who were surveyed using stratified random sampling. The results indicate that the adoption of online databases in student learning is influenced by six determinants, namely: (i) perceived effectiveness, (ii) perceived ease of use, (iii) technical barriers, (iv) personal usefulness, (v) usage attitudes, and (vi) convenience. Our study has revealed that students' intention to use the online database system is positively influenced by their perceived ease of use and perceived usefulness. These findings could be valuable in shaping policies for enhancing the online database system at economics universities, taking into account the students' characteristics and the institution's needs.

**Keywords:** economics universities; learning; online database systems; students

## 1. Introduction

Technology has greatly impacted human life, particularly in the field of education. As a result, the development of online database systems at universities is rapidly increasing both globally [1] and in Vietnam. The United States, a leader in education, has over 80% of its universities developing their online database systems, according to Cyber Universities 2018 statistics. Similarly, Vietnamese universities have been developing online database systems, such as the learning management system (LMS) and online libraries, to support student learning.

Policymakers and service system developers from universities should identify the factors that affect student usage to effectively develop online databases. Whereas studies evaluating the factors influencing the use of online database systems are common worldwide, some notable studies have identified key factors affecting the success of the learning management system [2] and the use of academic online databases in higher education [3]. These studies highlight the importance of understanding the needs of both students and faculty members to effectively develop online database systems that meet their requirements. However, in the context of Vietnam, research on the use of online databases is still limited, as most Vietnamese universities have only recently begun developing these systems. At economics universities, the online database system has only been promoted since 2019 due to the COVID pandemic, and there has not been a study on the factors affecting the effectiveness of student use. Therefore, conducting systematic research is essential to improving the online database system in universities.

The remainder of this study is organized as follows: Section 2 reviews the previous studies on online database systems, including LMS, e-library, and e-learning. Section 3 describes the methodology used to collect the data sample. Section 4 presents the results

of the analysis and discusses potential solutions. Finally, Section 5 provides some key conclusions for practical applications and recommendations.

## 2. Literature Review

The purpose of identifying the most significant difficulties of high school students in using online databases and CDROMs is to propose design elements and instructional strategies to make these tools more valuable as learning resources and identify the most important issues related to the use of electronic information resources in schools.

Groote [4] conducted an outreach survey of 188 UIC Peoria faculty staff, residents, and students to assess the use of online journals, the use of print journals, the use of databases, the level of computer literacy, and other characteristics of library users. The conclusion is that users prefer online resources for printing, and many choose to access these online resources remotely. The result of the study also shows that the usability of electronic libraries is a factor promoting the use of users for this system, the convenience and availability of the entire text seem to play a role in the selection of online resources for users.

To study the use of academic online databases in education at the University of West Florida, a survey involving the use of faculty staff and perspectives towards online databases was conducted [3]. Most respondents ($n = 46$) felt fairly consistent with academic databases via the library at the university. However, some faculty members argue that databases such as updated figures and social science citation indicators should be proposed for future inclusion. Booker et al. [5] also carried out research for business students on the delivery of information literacy instruction (ILI) through the application of online library resources (OLR) by business students. Research using web-based surveys, including closed-ended and open-ended questions, was conducted on 337 business students. The analysis results based on the TAM theoretical model indicate that the ILI of students was only beneficial in the early stages of using the library's digital resources. This benefit would be reduced or very little in the final results of use. At Limkokwing University of Innovative Technology in Malaysia, a study of the factors influencing the success of LMS was conducted by Jafari et al. [2]. The research model was developed by examining the relationship between student outcomes (perceived usefulness) and information quality, system quality, and readiness for online learning through the use of systems and user satisfaction, quantitative data obtained through questionnaires. After analysis, the data indicated that all relationships from the independent variable to the dependent variable were significant, including (A) system quality, (B) information quality, (D) system use, (E) user satisfaction (F), and user-perceived usefulness, except for the relationship between readiness for e-learning and system use. The most influential variable is the quality of information about user satisfaction and perceived usefulness, and the least influential variables are readiness for online learning, system use, and perceived usefulness. At Bareyo University, a study of the factors affecting the use of electronic databases by academic staff was conducted by Farouk and Muhammad [6] with the aim of investigating the level of use, the enabling factors, and the factors that impede the use of electronic databases in the university's library. The study uses a descriptive statistical approach to analyze the data collected and offers factors that facilitate the use of the database, some of which include: readiness to adapt to change, the availability of computers and ICT skills, internet access, management support, and the awareness of the user's initial electronic database. However, the cost of accessing and using online databases, infrequent power supplies, a lack of awareness, and too many difficult-to-remember passwords were found to be obstacles to the use of electronic databases. In another study, Chen et al. [7] launched a database examination study using structural equation modeling and Rasch modeling to explore the contributing factors of learning and research in higher education from a psychological assessment perspective. The study used ODAS modeling and feedback analysis of 300 graduate students in Shanghai, collected using the stratified random sample technique. The results showed that graduate students' usefulness and ease of use of the

database played an intermediate role in establishing a connection between self-efficacy and their intention to use computers and satisfaction with the database for research and learning. In addition, the results of the analysis show that student satisfaction is indirectly explained by the usefulness of the database through ease of use and intention to use.

Regarding the domestic database system, domestic studies mainly focus on specific types of databases. It is possible to mention the research on factors affecting the intention to use the e-learning system of students: the case of Hanoi University of Technology [8]. This research has provided useful suggestions for policymakers [9] and developers of e-learning systems at the economics universities, such as: towards the core interests of learners; building friendly systems; and improving technical barriers. In addition, the factors affecting students' intention to use mobile applications for education in Vietnam have contributed three basic meanings to science and practice: the suitability of the research model based on the TAM model; factors affecting the intention to use mobile applications for education; and as a reference source for related studies [10]. At the same time, it offers solutions to enhance applications such as investing in research and design; adding features and experiences; solving technical barriers; and improving application efficiency. It is impossible not to mention the research on the factors affecting the intention to use e-libraries of students at universities in Hanoi based on the TAM model to conclude whether the electronic library provides information and documents on many different topics, covering a long time and whether regular and timely updates will help students' learning, helping to increase the intention to use electronic libraries. At the same time, they also recommend promoting communication about electronic libraries to students, pointing out weaknesses to overcome [11].

Previous research has primarily focused on specific database subjects such as LMS, e-library, and e-learning, but there is a lack of research related to the online database system in general, especially in the context of economics universities in Vietnam. Therefore, this study aimed to investigate the determinants that impact the use of the online database system in the learning of students at six economics universities. The study provides recommendations to improve the efficiency of the online database system and to better meet the needs of students in the university. The study used three theories and six hypotheses, which are shown below:

The theory of reasoned action (TRA) explains consumer behavior and determines their behavioral predisposition based on general feelings of liking or disliking, which lead to behavior, and subjective norms, which refer to the influence of others on their attitudes [12].

The technology acceptance model (TAM) is a theoretical model used to evaluate the effects on the choice behavior of using technological devices for individual or collective needs [13].

The theory of planned behavior (TPB) is a theory developed from the TRA, which assumes that a behavior can be predicted or explained by behavioral tendencies to perform that behavior. Behavioral tendencies include motivational factors that influence behavior and are defined as the degree of effort that people put into the behavior [14].

Perceived effectiveness (PE) was defined as the personal perception of an individual's ability to use the system effectively. Individuals believe that their ability to use the system will impact the expectations and usefulness of the service and promote their intention to use the system. It was affirmed that perceived effectiveness had a significant influence on the perceived usefulness and intention to use the E-learning system of students at Hanoi University of Science and Technology [8].

**Hypothesis H1a:** *The perceived effectiveness of the online database system has a positive impact on students' intention to use it for learning purposes.*

**Hypothesis H1b:** *Perceived effectiveness positively impacts the perceived usefulness of the online database system.*

Perceived ease of use (PEU): According to research by Le and Dao [8], perceived ease of use is the perception of the ability to easily use the service when individuals are exposed to the service system. The studies of Taylor and Todd [15]; Venkatesh and Davis [16]; and Saroia and Gao [17] showed the synergistic effect of perceived ease of use on perceived usefulness on the indirect intention of use. Perceived ease of use helps users have a happy attitude and enjoy using services and products, thereby improving the intention to use [18].

**Hypothesis H2a:** *Perceived ease of use positively influences the perceived usefulness of the online database system.*

**Hypothesis H2b:** *Perceived ease of use has a positive impact on the attitude towards using the online database system.*

Technical barriers (TB): Based on the research theory of Julander and Soderlund [19], technical barriers are the disadvantages in terms of the technology and techniques to access the service system. Technical barriers have a major impact on the system acceptance process. Therefore, to develop the system effectively, technological and technical factors must be focused on reducing technical barriers for users [8].

**Hypothesis H3:** *Technical barriers negatively affect students' intention to use the online database system for learning purposes.*

Perceived usefulness (PU): Perceived usefulness is the degree to which users feel that using the system will help them improve their efficiency at work. In addition, the usefulness of the service is shown by helping customers save time, costs, and access to diverse services [20]. Studies by Kim et al. [21] concluded that the greater the perceived usefulness of users, the greater the influence on the intention to use.

**Hypothesis H4:** *Perceived usefulness has a positive influence on students' intention to use the online database system for learning purposes.*

Attitude to use (ATU): According to Ajzen [14], the intention to use something is directly influenced by "attitude", "subjective norm", and "behavioral control perception". Attitude is defined as an individual's positive or negative emotions when performing a behavior with a clear purpose [22]. When the individual has a positive attitude toward an action, the likelihood of performing that action is higher [23].

**Hypothesis H5:** *A positive attitude towards using the online database system has a positive impact on students' intention to use it for learning purposes.*

Convenience (CE): According to mental computation theory, convenience means consuming less physical and mental energy to reduce time and effort to increase the benefits of activities. For services or technology products, convenience is also the ability to access and use the service system, which the service system provides to users [24]. In a study by Gupta and Kim [25], it was shown that the convenience of the service also boosts the user's intention to use the service system.

**Hypothesis H6:** *Convenience has a positive influence on students' intention to use the online database system for learning purposes.*

Based on the theory of previous research, the study combined the analysis and determination of gaps in research. The research model has been designed in Figure 1.

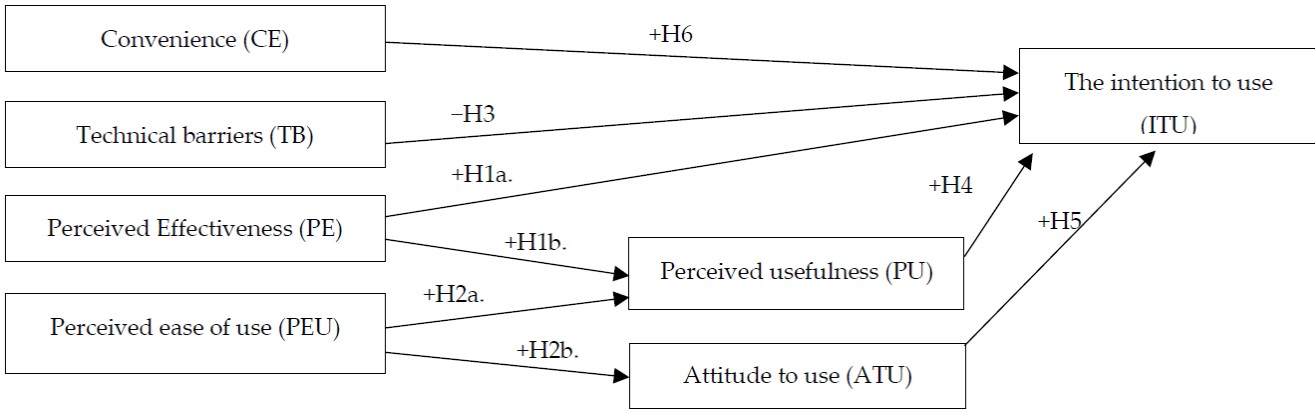

**Figure 1.** Proposed research model.

## 3. Methodology

The sample consisted of 492 students from six economics universities in Vietnam, selected through stratified random sampling. These six universities have different factors in terms of investment in facilities, students' admission scores, and different faculty staff. This will ensure the objectivity and comprehensiveness of the survey as well as the high reliability of the research results on student satisfaction when using online databases in learning. The survey was fully accepted by the survey participants, and students agreed to answer and provide information for the study. After the survey, 492 quality questionnaires were filtered and used to conduct the survey, then the data were encoded, entered, and cleaned.

The data were collected through a questionnaire containing 29 observed measurement variables for seven proposed groups of factors, including six independent factors and one dependent factor. The study focused on analyzing and synthesizing six concepts and problems related to the use of database systems. These concepts and problems were selected from various studies, including perceived ease of use and perceived usefulness [20,26–30], perceived effectiveness [28,31], convenience [24,25], attitude to use [14,22,23], and technical barriers [8,19].

Based on the theoretical models of the theory of reasoned action (TRA), technology acceptance model (TAM), and theory of planned behavior (TPB), and the results of the research by Le and Dao [8], the study proposed a theoretical model that identified factors affecting the use of online database systems in the learning of students at economics universities in Vietnam. The influencing factors were measured by six factors: perceived effectiveness, perceived ease of use, technical barriers, perceived usefulness, attitude to use, and convenience.

Survey Questions: The author of this study developed a questionnaire consisting of 7 factors and 29 observations (Appendix A) based on previous studies, building a research model, developing hypotheses, and performing testing, regression, and evaluation analysis for results and solutions. The questionnaire is developed to be suitable for students at universities in Vietnam. The survey respondents fully agreed to participate and provide information for the study. The questionnaire consists of 29 variables, rated on a Likert scale from 1 to 5 (level 1: strongly disagree. Level 2: disagree; Level 3: normal; Level 4: agree; Level 5: strongly agree) to measure the influence of factors affecting the use of online database systems in the learning of students at economics universities. In the questions, the database systems mentioned are all databases of the University of Economics, including LMS (learning management system), e-Learning, the system of websites, e-library, Ebooks, Facebook, and Zalo that students are using.

In quantitative research, the data analysis used for this study are exploratory factor analysis (EFA), confirmatory factor analysis (CFA), and structural equation modeling (SEM). When conducting exploratory factor analysis (EFA), the sample size should be at least 50,

preferably 100, and the observation/measurement ratio should be at least 5:1 and preferably 10:1 according to Hair et al. [32] and Nguyen [33]. In this study, the total number of observed variables is 29 observed variables for 7 proposed groups of factors. The research scale for the proposed factor groups is inherited from the study of Jalilvand et al. [34], so the minimum sample size is $29 \times 5 = 145$ and the best sample size is $29 \times 10 = 290$. However, because the rate of qualitative response to survey questionnaires is usually not high, in order to ensure the minimum sample size, the study surveyed the questionnaires and collected 554 survey questionnaires after reviewing and filtering out 492 votes. The larger the sample size, the smaller the error, and the higher the reliability of the research results. In this research, the sample size was chosen as $n = 492$. The software used for the statistical analysis is SPSS and AMOS.

## 4. Results

### 4.1. Scale Reliability Evaluation

The reliability of the scale was assessed using Cronbach's Alpha coefficient. The results of the Cronbach's Alpha test for variable groups in the study model show that:

The concept of perceived effectiveness (PE) is measured by three component scales with a Cronbach's Alpha reliability value of 0.763, achieving a good value for the unidirectionality of each component of the scale, in which the total correlation coefficients fluctuate from the lowest at 0.583 to the highest at 0.611; the alternative reliability coefficient when the highest scale component type is 0.695, compared to the measured Cronbach's Alpha value of 0.763, which is still smaller. Therefore, the scale of components of this concept is conditioned on measuring aspects of the concept.

The concept of perceived ease of use (PEU) is measured by five component scales with a Cronbach's Alpha reliability value of 0.851, achieving a good value for the unidirectionality of each component of the scale, in which the correlation coefficients vary from the lowest 0.635 to the highest 0.692, the reliability coefficients replace when the highest scale component type is 0.828, compared to the measured Cronbach's Alpha value of 0.851, which is still smaller. Therefore, the scale of components of this concept is conditioned on measuring aspects of the concept.

The concept of technical barriers (TB) is measured by four component scales with a Cronbach's Alpha reliability value of 0.805, achieving a good value for the unidirectionality of each component of the scale, in which the total correlation coefficients fluctuate from the lowest 0.602 to the highest 0.643; the alternative reliability coefficients when the highest scale component type is 0.828, compared to the measured Cronbach's Alpha value of 0.764, which is still smaller. Therefore, the scale of components of this concept is conditioned on measuring aspects of the concept.

The concept of perceived usefulness (PU) is measured by four component scales with a Cronbach's Alpha reliability value of 0.802, achieving a good value for the unidirectionality of each component of the scale, in which the correlation coefficients vary from the lowest 0.602 to the highest 0.645; the reliability coefficient replace when the highest scale component type is 0.759, compared to the measured Cronbach's Alpha value of 0.802, which is still smaller. Therefore, the scale of components of this concept is conditioned on measuring aspects of the concept.

Based on the test results in Tables 1 and 2, all 29 observed variables showed satisfactory results, indicating that the scale used in the implementation of EFA is reliable. Therefore, 29 observations are sufficient to ensure the reliability of the scale.

**Table 1.** Cronbach's Alpha Scale Reliability Test Results.

| Items | PEU | PE | TB | PU | ATU | CE | ITU | |
|---|---|---|---|---|---|---|---|---|
| Cronbach's Alpha | 0.851 | 0.763 | 0.805 | 0.802 | 0.785 | 0.837 | 0.866 | Total |
| Number of inspection observations | 05 | 03 | 04 | 04 | 03 | 05 | 05 | 29 |
| The number of observations accepted | 05 | 03 | 04 | 04 | 03 | 05 | 05 | 29 |
| Number of observations removed | 00 | 00 | 00 | 00 | 00 | 00 | 00 | 00 |

**Table 2.** Cronbach's Alpha and pattern matrix after extracting unmoderated items.

| Items | Factor | | | | | | | Cronbach's Alpha |
|---|---|---|---|---|---|---|---|---|
| | 1 | 2 | 3 | 4 | 5 | 6 | 7 | |
| Perceived ease of use (PEU) | | | | | | | | 0.851 |
| PEU4 | 0.862 | | | | | | | 0.813 |
| PEU5 | 0.767 | | | | | | | 0.825 |
| PEU2 | 0.709 | | | | | | | 0.815 |
| PEU1 | 0.645 | | | | | | | 0.822 |
| PEU3 | 0.625 | | | | | | | 0.828 |
| Convenience (CE) | | | | | | | | 0.837 |
| CE2 | | 0.786 | | | | | | 0.795 |
| CE3 | | 0.723 | | | | | | 0.802 |
| CE4 | | 0.720 | | | | | | 0.797 |
| CE1 | | 0.676 | | | | | | 0.809 |
| CE5 | | 0.609 | | | | | | 0.816 |
| The intention to use (ITU) | | | | | | | | 0.866 |
| ITU3 | | | 0.839 | | | | | 0.833 |
| ITU2 | | | 0.716 | | | | | 0.842 |
| ITU5 | | | 0.709 | | | | | 0.829 |
| ITU1 | | | 0.655 | | | | | 0.838 |
| ITU4 | | | 0.533 | | | | | 0.845 |
| Technical barriers (TB) | | | | | | | | 0.805 |
| TB3 | | | | 0.753 | | | | 0.761 |
| TB4 | | | | 0.736 | | | | 0.744 |
| TB1 | | | | 0.672 | | | | 0.753 |
| TB2 | | | | 0.619 | | | | 0.764 |
| Perceived usefulness (PU) | | | | | | | | 0.802 |
| PU3 | | | | | 0.748 | | | 0.738 |
| PU4 | | | | | 0.713 | | | 0.758 |
| PU1 | | | | | 0.695 | | | 0.759 |
| PU2 | | | | | 0.678 | | | 0.755 |
| Perceived effectiveness (PE) | | | | | | | | 0.763 |
| PE2 | | | | | | 0.732 | | 0.664 |
| PE1 | | | | | | 0.698 | | 0.688 |
| PE3 | | | | | | 0.680 | | 0.695 |

**Table 2.** *Cont.*

| Items | Factor | | | | | | | Cronbach's Alpha |
|---|---|---|---|---|---|---|---|---|
| | 1 | 2 | 3 | 4 | 5 | 6 | 7 | |
| | | | Attitude to use (ATU) | | | | | 0.785 |
| ATU3 | | | | | | | 0.736 | 0.680 |
| ATU1 | | | | | | | 0.701 | 0.715 |
| ATU2 | | | | | | | 0.665 | 0.731 |
| Kaiser–Meyer–Olkin (KMO) Measure of Sampling Adequacy | | | | | | | | 0.901 |
| Sig. (Bartlett's Test of Sphericity) | | | | | | | | 0.000 |
| Cumulative (%) | | | | | | | | 65.100 |
| The Value of Initial Eigenvalue | | | | | | | | 1.101 |

### 4.2. Exploratory Factor Analysis (EFA)

By PCA extraction and Promax rotation, the EFA test results of independent variables for KMO and Barlett's test results showed that KMO = 0.901 > 0.05 and Sig. = 0.000 < 0.05, thereby concluding that the observed variables included in the analysis are correlated with each other and the appropriate EFA discovery factor analysis used in this study in Table 3.

**Table 3.** Results of EFA exploratory factor analysis for independent variables.

| KMO and Bartlett's Test | | |
|---|---|---|
| Kaiser-Meyer-Olkin Measure of Sampling Adequacy | | 0.901 |
| Bartlett's Test of Sphericity | Approx. Chi-Square | 6253.089 |
| | df | 406 |
| | Sig. and nbsp | 0.000 |

The results of the factor analysis also show that the total variance explained is 65,100% > 50%, and the stopping point when deducting at the seventh factor is 2.722 > 1 all meet the conditions. Seven factors were drawn from the analysis of 29 included scales. The rotation matrix results of the EFA analysis show that seven new groups of factors with observed variables with factor load coefficients greater than 0.3 are satisfactory.

The study conducted EFA analysis with seven factors, PEU, CE, ITU, TB, PU, PE, and ATU, which were inherited from previous studies. The reason why this study used EFA is that these factors may be meaningfully relevant to previous studies when conducting a survey in this study about the use of online data in learning among students. The University of Economics in Vietnam may give different results, which are not necessarily consistent or similar to the results of previous studies. Therefore, the study uses EFA analysis with five-Likert scales to assess the subjective opinions of students at six universities of economics surveyed about the use of online databases in learning. Which factors and observed variables have the greatest impact and how do they have a linear relationship, correlated with each other? The results of the EFA analysis (Appendix B) indicate that the factor loading of all variables meets the requirements (>0.5), from which the seven groups were placed in the following order:

Group 1: The results of the EFA analysis on "Perceived ease of use" show that five measurement criteria are ranked by rising influence: PEU3, PEU1, PEU2, PEU5, and PEU4 with factor loadings from 0.625 to 0.862. Thus, the variable "You find it easy to exchange information with the online database system" has the most influence on the use of the online database system by students.

Group 2: The results of the EFA analysis on "Convenience" show that five measurement criteria are sorted by ascending influence: CE5, CE1, CE4, CE3, and CE2 with factor loadings from 0.609 to 0.786. The variable "The current online database is easily accessible" has the biggest impact.

Group 3: The results of the EFA analysis on "Intention to use" show that five measurement criteria are ranked by ascending influence: ITU4, ITU1, ITU5, ITU2, and ITU3 with factor loadings from 0.533 to 0.839. The variable "You will continue to use online databases that are in use for a long time and often" has the highest value and greatest influence.

Group 4: The results of the EFA analysis on "Technical barriers" show that four measurement criteria are sorted by ascending influencing degrees: TB2, TB1, TB4, and TB3 with factor loadings from 0.619 to 0.753. "Accessing online databases requires separate software, requiring an understanding of information technology of users" has the greatest value and impact.

Group 5: The results of the EFA analysis on "Perceived usefulness" show that four measurement criteria are arranged by ascending level: PU2, PU1, PU3, and PU4 with factor loadings from 0.678 to 0.748. "Use the online database to bring utilities" has the highest value.

Group 6: The results of the EFA analysis on "Perceived Effectiveness" show that three measurement criteria are sorted by increasing level of influence: PU3, PU1, and PU2 with factor loadings from 0.680 to 0.732. "You put your trust in using online databases for your learning" gives the greatest value.

Group 7: The results of the EFA analysis on "Attitude to use" show that three measurement criteria are arranged by ascending level of influence: ATU2, ATU1, and ATU3 with factor loadings from 0.665 to 0.736. "You feel that online databases have many advantages" gives the highest value.

### 4.3. Confirmatory Factor Analysis (CFA)

Look at the CFA results for the model in Table 4:

**Table 4.** Model fit results.

| CMIN/DF | CFI | TLI | RMSEA |
|---------|-----|-----|-------|
| 1.852 | 0.949 | 0.942 | 0.042 |

First, because Chi-square/df = 1.852 < 2, TLI = 0.942 > 0.90, CFI = 0.949 > 0.90, and RMSEA = 0.042 < 0.08, it can be said that the model is suitable for the market data.

Second, the (normalized) weights are all greater than 0.5. It ranges from 0.666 to 0.797, and all have $p < 0.05$, so the scales reach the convergence value.

Third, because the model is consistent with market data and the observed variables are not correlated, the scale achieves unidirectionality based on indicators: AVE > 0.5 and CR > 0.7.

Fourth, the AVE coefficients of the above seven groups are all larger than MSV, so the scale achieves differentiation. Thus, the model is consistent with market data, the concepts of achieving convergent value, achieving unidirectionality, distinguishing value, and measuring scale reliability in Table 5.

**Table 5.** CR and AVE evaluation table.

|  | CR | AVE | MSV | MaxR(H) | CE | PEU | TB | ITU | ATU | PU | PE |
|---|---|---|---|---|---|---|---|---|---|---|---|
| CE | 0.838 | 0.51 | 0.366 | 0.84 | 0.714 |  |  |  |  |  |  |
| PEU | 0.852 | 0.535 | 0.306 | 0.853 | 0.343 | 0.732 |  |  |  |  |  |
| TB | 0.805 | 0.509 | 0.356 | 0.807 | 0.391 | 0.372 | 0.713 |  |  |  |  |
| ITU | 0.866 | 0.565 | 0.402 | 0.868 | 0.605 | 0.553 | 0.597 | 0.751 |  |  |  |
| ATU | 0.786 | 0.551 | 0.362 | 0.792 | 0.559 | 0.199 | 0.537 | 0.602 | 0.742 |  |  |
| PU | 0.802 | 0.504 | 0.133 | 0.804 | 0.364 | 0.269 | 0.197 | 0.363 | 0.169 | 0.71 |  |
| PE | 0.764 | 0.519 | 0.402 | 0.766 | 0.313 | 0.509 | 0.499 | 0.634 | 0.434 | 0.229 | 0.72 |

### 4.4. Structural Equation Modeling Analysis (SEM)

According to the following summary table, in theory for good, the three indicators GFI, TLI, and CFI are all above 0.9. However, it is difficult to achieve all three indicators. According to Nguyen and Nguyen [35], the model with TLI, CFI $\geq$ 0.9 and CMIN/df $\leq$ 3, and MRSEA $\leq$ 0.08 is acceptable. Take this criterion and compare it with the actual acceptable results of the data set by the model.

The SEM model analysis results show that the research model is consistent with market data: Chi-square; CMIN/df = 2.362; GFI = 0.897; TLI = 0.908; CFI = 0.918; RMSEA = 0.053 in Figure 2.

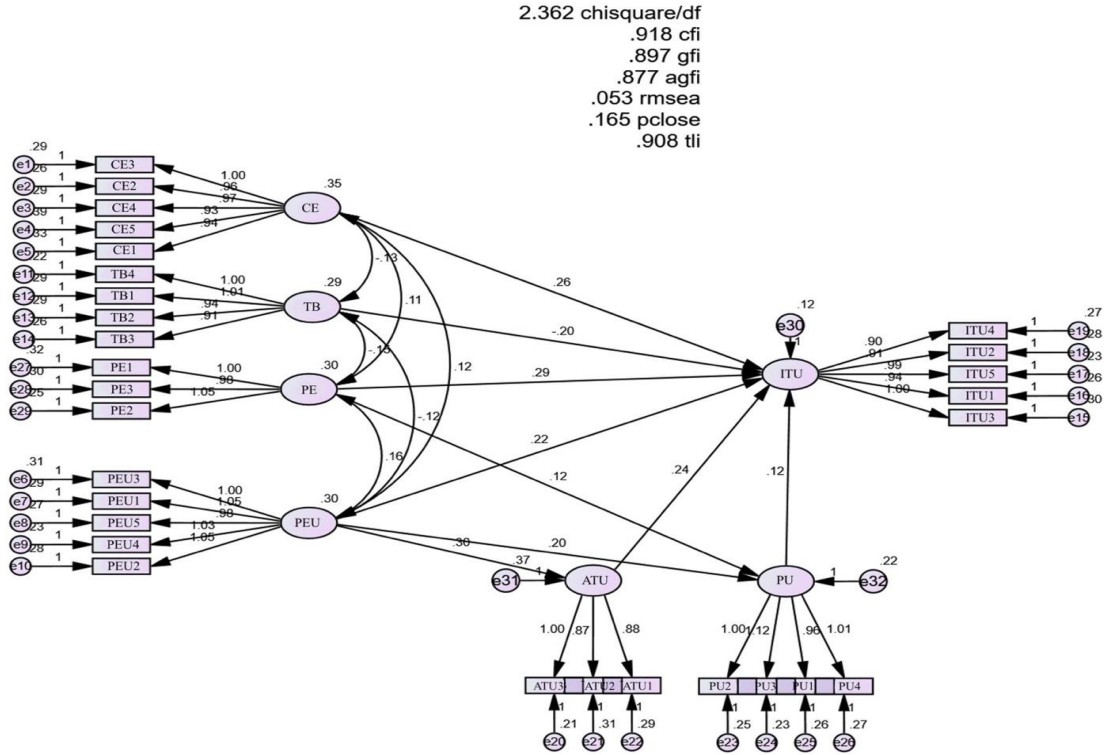

**Figure 2.** SEM analysis.

Table 6 shows the results of the hypothesis testing for the relationships between the factors in the proposed model. The *p*-values for the relationships are all below the 5% significance level. The relationship between PE and PU has a reliability of 90% at the 5% significance level, indicating that PE does not have a significant impact on PU in the same direction. However, at the 10% significance level, PE has a significant positive impact on PU. The study can also observe that at $p < 0.05$, PE, PEU, CE, ATU, and PU have a positive impact on ITU, while TB has a negative impact on ITU.

**Table 6.** Results of SEM model estimation.

| Relationship | | | Unstandardized | Normalizations | Standard Error | *p*-Value |
|---|---|---|---|---|---|---|
| ATU | <--- | PEU | 0.296 | 0.259 | 0.063 | 0.000 |
| PU | <--- | PEU | 0.196 | 0.216 | 0.061 | 0.001 |
| PEU | <--- | PE | 0.116 | 0.129 | 0.063 | 0.064 |
| ITU | <--- | PE | 0.293 | 0.273 | 0.063 | 0.000 |
| ITU | <--- | PEU | 0.223 | 0.206 | 0.057 | 0.000 |
| ITU | <--- | TB | -0.196 | −0.179 | 0.056 | 0.000 |
| ITU | <--- | CE | 0.257 | 0.256 | 0.047 | 0.000 |
| ITU | <--- | ATU | 0.235 | 0.249 | 0.041 | 0.000 |
| ITU | <--- | PU | 0.123 | 0.103 | 0.05 | 0.013 |

## 5. Discussion

The survey data analysis showed that all six factors examined (convenience, technical barriers, perceived effectiveness, perceived ease of use, perceived usefulness, and attitude to use) significantly affected the use of online database systems by students at six economics universities in Vietnam. The factor with the strongest influence was perceived effectiveness, while perceived usefulness had a reverse effect.

These findings support the reliability of the technology acceptance model (TAM) in predicting students' intention to use online database systems, specifically the factors of perceived ease of use and perceived usefulness. This is consistent with recent studies in online learning systems and e-commerce services [8,26,27,36,37].

Furthermore, our study revealed the significant influence of technical barriers on students' intention to use online database systems. This emphasizes the importance of considering technology-related factors and implementing technological solutions to reduce barriers to users caused by technical issues. To improve the use of online database systems in student learning at economics universities.

The factor "Perceived effectiveness" with the strongest influence among the six elements of the model was perceived effectiveness, accounting for 27.3% of the variance in students' intention to use the online database system. This indicates that students at the university prioritize and expect features that provide value and direct benefits in their use of the system for learning. The desire for efficiency when using technology is likely to continue to grow as technology advances.

The results of the study suggest that perceived ease of use has a relatively strong influence on students' use of online database systems in their learning at six economics universities, with a percentage of 20.6%. The analysis indicates that students are more likely to adopt and utilize the database system if it has a user-friendly design. With self-study becoming increasingly popular, the need to find relevant documents is also increasing, making it crucial for the database system to be easily accessible and navigable for students.

Through the evaluation according to the intended behavior theory (TPB) of Fishbein and Ajzen [12], combined with the research results, it was found that the technical barrier strongly influenced, −17.9%, the use of the online database system in the learning of students of National Economics University. The barriers are easy to encounter such as incompatibility with equipment, software errors, etc. If this problem is not overcome, students will notice the inconvenience and reduce the use of the online database system in their learning.

From the analysis, it was found that the perceived usefulness has a relative impact on the use of the online database system in the learning of students at six economics universities, at 10.3%. Due to the popularity of the internet in Vietnam, we have more and more opportunities to access information and data, but not all sources of information are

useful to users. Therefore, the more valuable the database system is to students, the more students will use the system in the learning process.

The attitude towards using impacts on the use of online database systems in the learning of students at economics universities at a high level of 24.9%. Attitude is also affected by perceived ease of use at 25.9%. The perceived usefulness factor does not affect the use attitude factor. The DATA base system with proximity and convenience in use will be well received by students and have a good attitude to influence in the same direction as their use.

The research tested the hypothesis H6, and the results showed that the correlation between these two factors is relatively large, with a rate of 25.6%. The convenience of the system will promote the use of students because students always want to access and use the school's system easily.

Thus, the results show that six out of six factors affect the use of online database systems in the learning of students of six Vietnamese economics universities including convenience, technical barriers, perceived effectiveness, perceived ease of use, perceived usefulness, and attitude to use. This is consistent with previous studies. Firstly, the perceived effectiveness and convenience positively affect the use of the online database system in students' learning. This is consistent with research in Korea [28,31]. Second, research shows that the factors of the TAM model (perceived ease of use, perceived usefulness) are factors that positively affect students' intended use of the database system. This again validates the reliability of the TAM model for technology services. This result is quite similar to recent studies such as Le and Dao [8], Cakir and Solak [26] and Mohammadi [27] for online training systems or in e-commerce services such as Klopping and Mckinney [36] and Uroso et al. [37]. Third, the study found that attitudes to use positively impact the use of online database systems in students' learning. This was also mentioned in some previous studies by Davis [20] and Venkatesh and Davis [16]. Through the survey data analysis results, the study also noted the clear influence of technical barriers on the current E-learning system acceptance process. This is consistent with Julander and Soderlund [19] and Le and Dao [8]. If the system has technical barriers such as being difficult to use, incompatible with other devices or software, or does not meet performance or scalability requirements, the user may have difficulty using the system and may have a negative attitude or no intention of using it. Technical barriers can also affect the reliability and availability of the database system, causing problems or loss of data and compromising user trust in the system.

## 6. Conclusions and Recommendation

The study suggests implementing the following solutions:

### 6.1. For Economics Universities

Firstly, Vietnamese universities are not real service companies [33]. Therefore, economics universities need to improve the accessibility of the system to make it more convenient for students to access the system. The school needs to disseminate the general knowledge and benefits of the university's online database system to students by organizing seminars and workshops to guide new students on how to use the system so that students can better understand the system and be able to use all the functions and tools of the system optimally.

Secondly, the perceived effectiveness for students also needs to be improved through the establishment of a system that is compatible with all operating systems in students' devices, especially the need to install more system versions suitable for devices running macOS 11.0 or lower, build tools in the system friendly, regularly add new utilities as well as new features and experiences.

Thirdly, the school needs to improve the usefulness of the system. Students today have a very high demand for finding materials online, so universities should invest in and update a variety of learning materials on the Reader electronic library. In addition, lecturers

should also post more soft copies of documents in addition to the main material on the class in the LMS channel so that students can easily receive more sources of information and knowledge.

Fourthly, the online database system needs to be improved through a reduction in technical barriers. The school should invest in and upgrade the hardware of the system to avoid overloading and crashing the system every time there is a large number of students accessing it in a period of time.

Last, the school needs to strictly manage and control the system to promptly prevent hackers from hacking to disrupt the system or hackers posting unhealthy content that affects the school and students.

*6.2. For the Students*

Every student, especially freshmen, needs to actively register and participate in lessons, training seminars, and tutorials using the universities' online database system, thereby equipping themselves with the knowledge and skills necessary to use the channels of the support system for learning most effectively.

In addition, each student should actively update new versions of the application in the online database system on his/her devices to gain a better experience during use for his/her learning. Students should take advantage of these utilities and arrange and use the channels of the system for learning in a reasonable, effective way, avoiding wasting time.

Furthermore, the students can exchange and share the usage and access to their effective system. Students also regularly contribute comments to economics universities about the inadequacies and limitations of the system in the process of using it for learning so that the universities can promptly fix and repair it to bring the best experience to students.

The limitations of the study: The study only studied six influencers inherited from the overview of previous studies to the use of online databases of six representative universities in Vietnam. Other factors related to or arising from the research have not been studied. The study stopped at the study of the impact of the use of online data systems in the learning of students in terms of effectiveness, ease of use, technical barriers, usefulness, attitude to use, and convenience in using online data systems in the learning of students.

**Funding:** This research was funded by the National Economics University, Hanoi, Vietnam.

**Institutional Review Board Statement:** Not applicable.

**Informed Consent Statement:** Written informed consent was obtained from all subjects involved in the study.

**Data Availability Statement:** Data supporting the reported results can be requested from the first author.

**Conflicts of Interest:** The author declares no conflict of interest.

**Appendix A. Summary of Influencing Factors and 29 Observed Variables in the Survey Questionnaire**

| Content of Question | Reference |
| --- | --- |
| **1.** Perceived effectiveness (PE) | |
| You can easily use the online database. | [8,18,31] |
| You put your trust in using online databases for your learning. | |
| You are proficient in using electronic devices to access online databases. | |
| **2.** Perceived ease of use (PEU) | |
| You find the online database system easy to use. | [11,20,37] |
| Online database system makes it easy for users to learn. | |

| Content of Question | Reference |
|---|---|
| Easy online database system can be proficient in using. | |
| You find it easy to exchange information with the online database system. | |
| The online database system has a convenient interface for use. | |
| *3.* Technical barriers (TB) | |
| Some operating systems on electronic devices do not support online database system access. | |
| The ability to store the database and the speed of user access is affected by the information technology system. | [3,20,21] |
| Accessing online databases requires separate software, requiring an understanding of information technology of users. | |
| *4.* Perceived usefulness (PU) | |
| Using online databases to improve learning. | |
| Using online databases to improve learning outcomes. | [11,20,37] |
| Materials provided on the online database are useful to students. | |
| Use the online database to bring utilities. | |
| *5.* Attitude to use (ATU) | [9,20] |
| You prefer online databases. | |
| You like to use online databases for learning. | |
| You feel that online databases have many advantages. | |
| *6.* Convenience (CE) | |
| Online databases are accessible anytime, anywhere as long as there is an internet connection. | |
| The current online database is easily accessible. | [24,25] |
| Online database helps you save time studying. | |
| Online database helps you be proactive in arranging study time and finding documents. | |
| *7.* Intention to use (ITU) | |
| You want to experience all the features of the online database. | [9,20] |
| You are ready to experience and use new online databases. | |
| You will continue to use online databases that are in use for a long time and often. | |
| You will recommend these online databases to those who do not know or new students to the school. | |

## Appendix B. Summary of Research Results EFA

| | Components | | | | | | |
|---|---|---|---|---|---|---|---|
| | 1 | 2 | 3 | 4 | 5 | 6 | 7 |
| PEU4 | 0.862 | | | | | | |
| PEU5 | 0.767 | | | | | | |
| PEU2 | 0.709 | | | | | | |
| PEU1 | 0.645 | | | | | | |
| PEU3 | 0.625 | | | | | | |
| CE2 | | 0.786 | | | | | |
| CE3 | | 0.723 | | | | | |
| CE4 | | 0.720 | | | | | |

| | Components | | | | | | |
|---|---|---|---|---|---|---|---|
| | 1 | 2 | 3 | 4 | 5 | 6 | 7 |
| CE1 | | 0.676 | | | | | |
| CE5 | | 0.609 | | | | | |
| ITU3 | | | 0.839 | | | | |
| ITU2 | | | 0.716 | | | | |
| ITU5 | | | 0.709 | | | | |
| ITU1 | | | 0.655 | | | | |
| ITU4 | | | 0.533 | | | | |
| TB3 | | | | 0.753 | | | |
| TB4 | | | | 0.736 | | | |
| TB1 | | | | 0.672 | | | |
| TB2 | | | | 0.619 | | | |
| PU3 | | | | | 0.748 | | |
| PU4 | | | | | 0.713 | | |
| PU1 | | | | | 0.695 | | |
| PU2 | | | | | 0.678 | | |
| PE2 | | | | | | 0.732 | |
| PE1 | | | | | | 0.698 | |
| PE3 | | | | | | 0.680 | |
| ATU3 | | | | | | | 0.736 |
| ATU1 | | | | | | | 0.701 |
| ATU2 | | | | | | | 0.665 |
| KMO | 0.901 | | | | | | |
| Eigenvalue | 1.101 | | | | | | |
| SigiBarlett | 0.000 | | | | | | |
| Total variance extracted | 65.100 | | | | | | |

Source: Compiled from SPSS analysis.

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
