# Peer review of "Factors Affecting the Adoption of Online Database Systems for Learning among Students at Economics Universities in Vietnam"

_ejihpe, doi:10.3390/ejihpe13050062_

Round 1
Reviewer 1 Report
1. Results
a) The author refers to the fact that inappropriate responses were excluded (lines 238-240), but the sample corresponds to the number of questionnaires analysed. It is not clear what was excluded.
b) It would be interesting to explain if significant differences were identified considering the university attended. Why select six different institutions? These results could robust the study.
2. References
a) The following references are missing in the final:
Shahzad and Lodhi, 2020 (line 23)
Groote and Dorsch (2003) (line 58) – the reference is missing (final), only Groote (2003)
Le and Dao, 2015 (line 161/164/174/194)
Le, 2016 (line 179) – the reference is missing (final) or Le and Dao?
Erk and Evans, 2016 (line 200)
Hsu, 2016 (line 208) – the reference is missing (final) or Chou and Hsu?
Nguyen and Nguyen, 2008 (line 280)
Nguyen, 2013 (line 367)
b) Reference not used in the article
Reference 1 (lines 421-413)
c) The author uses, in the text, the system (author, date) and not [number]
d) Reference is not in proper alphabetical order:
Uroso et al., 2010 (line 306)/Reference (line 457)
Author Response
Dear Editor,
Thank you so much for giving valuable comments for me to revise my paper. Absolutely, after revising the paper, the quality is better.
Best Regards!

Reviewer 2 Report
1- Citation is an issue here. For example "(Booker et al., 2012) also carry out research" should be "Booker et al. (2012) also carried out research".
2- Why did the authors combine between the theoretical framework and methodology. These should be two separate sections.
3- The hypotheses should not be in the methodology section, but in the introduction.
4- Table 2 in unclear. Attempts should be put to make it clear.
5- It is not clear why you used exploratory factor analysis.
6- You should use literature in the discussion section.
English language is an issue here. For example, "is conducted" (line 68) should be "was conducted". Another mistake is in "(Booker et al., 2012) also carry out research". Another mistake is in "(Booker et al., 2012) also carry out research", which should be "Booker et al. (2012) also carried out research".
Author Response
Dear Editor,
Thank you so much for giving valuable comments for me to revise my paper. Absolutely, after revising the paper, the quality is better. Please see the attachment.
Best Regards!

Reviewer 3 Report
Dear authors, the document you present is interesting and well constructed. However, I would like to suggest some areas for improvement:
- In the literature review, the first paragraph does not make much sense as it is from 1995 and is totally out of date with the reality in which we live.
- The section on the theoretical framework is decontextualised, as it only mentions theories without linking them or justifying why they are included in this study or their relevance to it.
- Regarding the questionnaire of 26 variables, it is necessary to say if you have developed it yourself, if it was already validated, and if so, the authorship of the same.
- The theoretical models, the objective of the research and the different hypotheses that you put forward should come before the methodology. This is a consequence of the literature review and the theoretical framework and not of the methodology.
- The methodology does not say when the study was carried out or whether the students' permission was obtained.
- In the sample it is stated that 492 were collected and that after the elimination of some incorrect ones, 492 were left.
- There is no justification why 145 is the minimum number of the sample.
- There is no mention of the software(s) used for the statistical analysis.
- The limitations of the study are missing.
Author Response
Dear Editor,
Thank you so much for giving valuable comments. I revised my paper. Please see the attachment.
Best Regards!

Round 2
Reviewer 2 Report
It is not clear why you use exploratory factor analysis with all the factors, where most of the factors were adapted from the literature. The author should explain this issue.
English language is still an issue here.
The language is still an issue here. In the following sentence three tenses are used.
Research using web-based surveys, including closed-ended and open-ended questions, was conducted for 337 business students. The analysis results based on the TAM theoretical model indicate that the ILI of students is only beneficial in the early stages of using the library's digital resources. This benefit will be reduced or very little in the final results of use. At Lim kokwing University of Innovative Technology in Malaysia, a study of the factors influencing the success of LMS was conducted by (Jafari et al., 2015).
Author Response
Dear Editor,
Thank you so much for your valuable comments about my paper. It's good for me to revise my paper. So, after revising, the paper is better quality. I highly appreciate your help. Please see file attached.
Best Regards.
Thi Minh Phuong Nguyen

Reviewer 3 Report
Thank you very much for your responses to my comments and suggestions. I have found that you have made the changes that were suggested and the work has improved significantly. Congratulations.
Best regards.
Author Response
Dear Editor,
Thank you so much for your valuable comments. It's useful for me to revise better quality paper.
I highly appreciate your help.
Best Regard!
Round 3
Reviewer 2 Report
Done
Would be better given to an English language editor.